# Cyclohumulanoid Sesquiterpenes from the Culture Broth of the Basidiomycetous Fungus *Daedaleopsis tricolor*
[note 1]

**DOI:** 10.3390/molecules26144364

**Published:** 2021-07-19

**Authors:** Ryuhi Kanehara, Akio Tonouchi, Katsuhiro Konno, Masaru Hashimoto

**Affiliations:** 1Faculty of Agriculture and Life Science, Hirosaki University, 3-Bunkyo-cho, Hirosaki 036-8561, Japan; h18a2013@hirosaki-u.ac.jp (R.K.); symbio@hirosaki-u.ac.jp (A.T.); 2Institute of Natural Medicine, University of Toyama, Toyama 930-0194, Japan; kkgon@inm.u-toyama.ac.jp

**Keywords:** cyclohumulanoid, DFT-based chemical shift calculation, DFT-based ECD spectral calculation

## Abstract

A series of cyclohumulanoids, i.e., tricocerapicanols A–C (**1a**–**1c**), tricoprotoilludenes A (**2a**) and B (**3**), tricosterpurol (**4**), and tricoilludins A–C (**5**–**7**) were isolated along with known violascensol (**2b**) and omphadiol (**8**) from the culture broth of *Daedaleopsis tricolor*, an inedible but not toxic mushroom. The structures were fully elucidated on the basis of NMR spectroscopic analysis, and the suggested relative structures were confirmed via density functional theory (DFT)-based chemical shift calculations involving a DP4 probability analysis. In the present study, the ^1^H chemical shifts were more informative than the ^13^C chemical shifts to distinguish the diastereomers at C-11. The absolute configurations of **1**–**5** were determined by comparing the experimental and calculated electronic circular dichroism (ECD) spectra. For **6** and **7**, the same chirality was assigned according to their biosynthetic similarities with the other compounds. The successful assignment of some Cotton effects was achieved by utilizing DFT calculations using simple model compounds. The plausible biosynthesis of **1**–**7** was also discussed on the basis of the structural commonality and general cyclohumulanoid biosynthesis. Compounds **2a** and **5** were found to simultaneously induce hyphal swelling and branching at 5.0 μg/mL against a test fungus *Cochliobolus miyabeanus.*

## 1. Introduction

Cyclohumulanoids belong to a family of polycyclic sesquiterpenes including illudanes, protoilludanes, suterpuranes, and africanes [1,2]. These compounds are biosynthetically related to humulene, a monocyclic 11-membered sesquiterpene, and are metabolites of plants, fungi, ferns, and marine sponges. In vitro biomimetic cyclizations of humulene conducted in the 1980s, mainly by Shirahama, yielded a series of cyclohumulanoids [3,4,5,6,7]. Modern biosynthetic studies have revealed the enzymes responsible for these processes and the detailed mechanisms [8,9,10]. To date, more than 150 members of this family have been reported [1,7], some of which exhibit notable biological activities. For example, ptaquiloside, a glycosylated illudoid of bracken, acts as an inducer of cancer [11], whereas irofulven, a derivative of illudin [12,13], is under clinical examination as an anticancer drug [14]. Moreover, 5-*O*-aryl esters of protoilludens exhibit antifungal activity and inhibit plant growth in lettuce [15]. The unique structures of this family of compounds have attracted the interest of synthetic chemists, resulting in numerous reports on their synthesis [16,17,18,19,20,21]. 

*Daedaleopsis tricolor* (*D*. *tricolor*) is one of the common basidiomycetes, wood-rotting fungi found in Europe [22,23,24,25] and Asia [26,27,28]. This fungus is not toxic but generally inedible. Interestingly, extensive investigations of its constituents have revealed various metabolites such as terpenoids and saccharides [25,26,27,28]. In the present study, we disclosed nine cyclohumulanoids from the culture broth of this fungus collected in the Shirakami mountainous area (Sirakami-Sanchi) of Japan, which UNESCO designated as a World Heritage Site in 1993 [29].

## 2. Results

*D*. *tricolor* sampled in 2019 at the Shirakami Natural Science Park of Hirosaki University located in the Shirakami Mountains, Japan, was cultured in potato dextrose medium for 2 weeks under shaking conditions. The ethyl acetate extracts from both the culture broth and the fungus body were subjected to a series of chromatographic separations to obtain tricocerapicanols A–C (**1a**–**1c**), tricoprotoilludenes A (**2a**) and B (**3**), tricosterpurol (**4**), tricoilludins A–C (**5**–**7**), and known violascensol (**2b**) [30] and omphadiol (**8**) as shown in Figure 1 [31]. Table 1 summarizes the ^1^H and ^13^C NMR data of the new compounds. The spectral data of known **2b** [30] and **8** [31] agreed with those reported in the literature. Although the molecular ion of **8** was not observed in the electron spray ionization–time of flight–mass spectrometry (ESI-TOFMS) analysis, its formula was confirmed after conversion into its 10-*O*-benzoate (**8-OBz**, Appendix A). Despite being addressed in various studies [30,32], the absolute configuration of **2b** was not fully established yet; this was successfully elucidated in the present study. The absolute configuration of **8** is discussed on the basis of our own investigations, although it was previously established synthetically [33]. 

The molecular ion of tricocerapicanol A (**1a**) was observed at *m/z* 237.1869 in the ESI-TOFMS spectrum (SI-6), suggesting that its molecular formula is C_15_H_24_O_2_ ([M + H]^+^: 237.1849). This was confirmed by the presence of 15 resonances in the ^13^C NMR spectrum. A 2D spectral analysis involving heteronuclear single quantum coherence (HSQC, SI-10) and heteronuclear multiple bond coherence (HMBC, SI-11) spectra suggested a cerapicane framework for **1a** [34,35]. Its ^1^H NMR spectral profile (SI-7) resembled that of repraesentin B, which was isolated by Makabe et al. in 2003 from the fungus *Lactarius repraesentaneus* [36]; however, there were some considerable differences, despite fact that CDCl_3_ was used as the solvent in both cases. For example, the H-2 of **1a** was observed at 2.27 ppm, whereas it appears at 2.37 ppm for repraesentin B. This chemical shift difference suggests that they are different compounds. The irradiation of the H-2 resonance of **1a** afforded nuclear Overhauser effects (NOEs) with Hβ-4, Hβ-5, H-9, and H_2_-12 as shown in Figure 2 (SI-12), revealing that **1a** is a C-11 epimer of repraesentin B. Other correlations observed in the 2D nuclear Overhauser effect spectroscopy (NOESY) spectrum not only confirmed the relative structure but also allowed assigning the prochiral methylene ^1^H signals, except for the conformationally flexible H_2_-12 (SI-13). Considering other novel compounds isolated in the present study, this compound was named tricocerapicanol A.

The ESI-TOFMS spectrum of tricocerapicanol B (**1b**) exhibited the molecular ion at *m*/*z* 253.1802 (SI-15). The mass difference from **1a** (Δ*m/z* 15.9933) indicated that **1b** contains one more oxygen than **1a**. The similar 2D spectral analysis as above allowed establishing that **1b** is a 5-hydroxylated derivative of **1a** (SI-18, 19, and 20). A NOESY correlation between H-5 and H-9 revealed the α-configuration for 5-OH (SI-21). Although the H-2 and H-9 signals appeared too close (2.28 and 2.35 ppm, respectively) to observe the NOESY correlation between them, these were assigned to take a *cis*-relationship because Hβ-4 afforded NOESY correlations with both H-2 and H-9. Another NOESY correlation between H-2 and H_2_-12 defined the relative configuration at C-11.

Tricocerapicanol C (**1c**) was found in the polar fraction eluted by silica gel column chromatography using 80% EtOAc/hexane. In the ^1^H NMR spectrum (SI-24), this molecule afforded two sets of broad AB doublets at the medium frequency region (3.14 and 3.20 ppm, *J* = 10.3 Hz (H_2_-12); 3.38 and 3.60 ppm, *J* = 11.0 Hz (H_2_-14)), suggesting the existence of two hydroxymethyl groups in the molecule. A 2D NMR spectral analysis (SI-26, 27, and 28) allowed us to conclude that **1c** is a C-14 hydroxylated congener of **1a**, which is in accordance with the observed molecular ion at *m/z* 253.1802 (C_15_H_25_O_3_^+^, [M + H]^+^: 253.1798) in the ESI-TOFMS spectrum (SI-23). The NOESY signals between H-2 and H-12, H-2 and Hβ-4, and Hβ-5 and H-9 were diagnostic of the relative structure depicted in Figure 1. Although the C-5 resonance (29.7 ppm) was overlapped with the solvent signal (acetone-*d*_6_, 29.9 ppm), this signal was assigned on the basis of the HSQC and HMBC spectra (SI-27 and 28, respectively). 

The ^1^H NMR spectral data of tricoprotoilludene A (**2a**, SI-32) resembled those of known protoilludane violascensol **2b** (SI-39), except for the absence of the H_2_-12 signal in **2a** [30]. The C-12 resonance was observed at 182.0 ppm, suggesting that **2a** is a carboxylic acid derivative of **2b**. This was confirmed by ESI-TOFMS spectrum (*m*/*z* 249.1482, C_15_H_21_O_3_^+^, [M + H]^+^: 249.1485, SI-31). Unfortunately, diagnostic NOEs of **2a** were not observed. Since the extracts also afforded **2b**, **2a** can be assumed to possess the same stereochemical relationship as **2b** by considering their biosynthesis. Hence, the prochiral methylene protons were tentatively assigned according to those of **2b**. The proposed relative configuration was supported by the theoretical ^13^C and ^1^H chemical shifts obtained by density functional theory (DFT) calculations, as will be described later.

The sodium adduct ion of tricoprotoilludene B (**3**) appeared at *m*/*z* 275.1620 in the ESI-TOFMS spectrum (SI-43), suggesting that **3** possesses the molecular formula C_15_H_24_O_3_ ([M + Na]^+^: 275.1618). The correlation spectroscopy (COSY) spectral data revealed the spin systems H_2_-1/H-2/C-9(/H_2_-8)/H_2_-10 and H_2_-4/H-5 (SI-46). The C-12 oxymethylene, C-13 methyl, and C-11 quaternary carbon atoms were assigned by the HMBC signals H_2_-12/C-1, H_2_-12/C-10, H_2_-12/C-11, H_3_-13/C-1, H_3_-13/C-10, H_3_-13/C-11, and H_3_-13/C-12 (SI-48). A W-coupling between Hβ-1 and Hβ-10 (1.4 Hz) not only supported the above assignment but also enabled the configurational discrimination of methylenes H_2_-1 and H_2_-10. The H_2_-15 and H_2_-8 signals showed HMBC correlations with C-6 and C-7 (146.1 and 134.1 ppm, respectively), and the C-6 resonance further correlated with H_2_-4, H-5, and H_3_-14. These results suggest that **3** possesses a protoilludane framework. The NOESY correlations between H-2 and H_2_-12, H-2 and H-5, Hα-4 and H_3_-14, Hβ-4 and H-5, and H-9 and H_2_-12 (SI-49) indicated that **3** has the same relative structure as **2b**. Meanwhile, the NOESY correlation between Hα-8 and H_3_-14 allowed distinguishing the prochiral H_2_-8 atoms.

Tricosterpurol (**4**) gave the largest ion at *m*/*z* 219.1760 indicating C_15_H_23_O^+^ (calcd. 219.1743) in the ESI-TOFMS spectrum (SI-51). This ion was tentatively assigned as the dehydrated ion because two oxygenated carbon signals were observed at 73.4 and 70.5 ppm (C-4 and C-12, respectively) in the ^13^C NMR spectrum (SI-53). Thus, C_15_H_24_O_2_ can be assigned as the molecular formula of **4**. The methylene protons H_2_-1 appeared at 2.07 and 2.25 ppm as broad AB doublets (SI-52), which showed HMBC correlations with the C-2 (140.1 ppm) and C-3 (126.6 ppm) sp^2^ carbons. The COSY spectrum indicated a long-range spin coupling between H_2_-1 and H_3_-14 (SI-54), although the signal splittings were not observed in the regular one-dimensional ^1^H NMR spectrum. These observations suggest a tetrasubstituted C-2/C-3 double bond. In the HMBC spectrum, the quaternary C-4 signal showed correlations with H_2_-5 and H_3_-14, and the other quaternary C-7 resonance (43.8 ppm) correlated with H_2_-6, H_2_-8, and H_3_-15, revealing the presence of a C-4/C-5/C-6/C-7 cyclobutane ring. This allowed us to assign a sterpurane framework for **4** [37]. The irradiation of H-9 (2.48 ppm) resulted in NOEs with Hβ-6 and H_2_-12, which unequivocally defined the relative configuration of **4** (SI-57). An analysis of the NOESY spectrum enabled the full assignment of the ^1^H and ^13^C NMR signals, except for the conformationally flexible prochiral H_2_-12 (SI-58).

Tricoilludin A (**5**) produced the largest ion at *m*/*z* 219.1738 in the ESI-TOFMS spectrum (SI-60), which corresponds to C_15_H_23_O^+^ (calcd. 219.1743). Similarly to **4**, the presence of two oxygenated carbon signals at 70.4 and 70.6 ppm in the ^13^C NMR spectrum indicates that this signal can be assigned to the dehydrated molecular ion (SI-62). Accordingly, the molecular formula of **4** was determined to be C_15_H_24_O_2_. The ^1^H NMR spectrum showed two characteristic sets of methylene proton signals at a low-frequency region (at 0.69 and 0.75 ppm (H_2_-4) and at 0.48 and 0.81 ppm (H_2_-5), SI-61) which showed the HMBC correlations with a quaternary carbon at 30.2 ppm (SI-65). The carbon atoms bonded to these protons resonated also at low frequency (5.7 and 7.7 ppm). These results reveal the presence of a spiro-cyclopropane ring (C-4/C-5/C-6) in the molecule. Since these cyclopropane methylene protons further correlated with quaternary carbons at 70.4 (C-7) and 125.2 ppm (C-3) in the HMBC spectrum, the cyclopropane ring can be assumed to be sandwiched between them. Further analysis of the 2D NMR spectra allowed establishing an illudane framework for **5**, [38] as illustrated in Figure 1. The irradiation of H-9 (2.55 ppm) resulted in NOEs with H_2_-12 and H_3_-15 (SI-66), which unequivocally defined the relative configuration. The presence of NOE signals between H*_R_*-4 and H_3_-14, H*_S_*-4 and H_3_-15, and H*_S_*-5 and H_3_-14 enabled distinguishing both ^1^H and ^13^C signals on the prochiral cyclopropane ring. However, like in other compounds, the prochiral methylene protons at C-12 could not be distinguished.

Tricoilludin B (**6**) afforded the sodium adduct ion at *m/z* 275.1623 along with the dehydrated ion at *m/z* 235.1700 in the ESI-TOFMS spectrum (SI-68), suggesting its molecular formula to be C_15_H_24_O_3_ ([M + Na]^+^: 275.1618, [M + H−H_2_O]^+^: 235.1693). This molecule also showed ethylene protons at smaller frequencies than 1.0 ppm in the ^1^H NMR spectrum (SI-69), which is consistent with an illudane framework. Notably, H-2 was newly observed at 2.32 ppm, and C-14 (71.2 ppm) was oxygenated. The HMBC correlation between Hα-14 (3.97 ppm) and C-7 (83.1 ppm) is indicative of an ether linkage between C-7 and C-14 (SI-73). This was supported by the acetylation of **6** under conventional conditions (Ac_2_O, pyridine) affording only a monoacetate (SI-76-80). Irradiation of Hα-14 resulted in an NOE with H_3_-13, which unequivocally defined the relative configuration of **6** (SI-74). A long-range spin coupling between H-2 and Hβ-14 (2.0 Hz, W-coupling) supported this configuration. The full assignment of the ^1^H and ^13^C NMR signals, except for the conformationally flexible H_2_-12 (SI-75), was achieved on the basis of a NOESY spectrum (SI-75).

The ESI-TOFMS spectrum of tricoilludin C (**7**) showed the molecular ion at *m*/*z* 253.1793 and the dehydrated ion at *m*/*z* 235.1689 (SI-82), suggesting the same molecular formula as that of **6** (C_15_H_24_O_3_, [M + H]^+^: 253.1798, [M + H−H_2_O]^+^: 235.1693). Furthermore, the ^1^H NMR spectra of both compounds were similar (SI-83). Nevertheless, a notable difference was found; one of the oxymethylene signals at 3.57 ppm (Hα-15) of **7** showed a long-range coupling with Hα-8 (1.70 ppm, ^4^*J*_Hα-8/Hα-15_ = 1.8 Hz), whereas a similar long-range coupling but between H-2 and Hβ-14 was observed for **6**. Accordingly, **7** contains a cyclic ether between C-3 and C-15 instead of the ether linkage between C-7 and C-14 in **6**. The HMBC spectrum supported the proposed relative structure (SI-87). The NOESY correlations observed between H-2 and H-9, H-2 and H_2_-12, and H-9 and Hβ-15 allowed establishing the relative configuration of **7** shown in Figure 2 (SI-88). Moreover, the NOESY signals between H*_R_*-4 and H_3_-14 and between H*_S_*-5 and Hα-1 enabled us to differentiate the prochiral H_2_-1, H_2_-4, and H_2_-5 methylene protons. 

The relative structures of **1a**–**7** and known **2b** and **8** were further investigated by DFT-based NMR chemical shift calculations because the variety of the framework of **1a**–**8** constituted interesting examples to evaluate the reliability of this methodology [39,40,41,42] and to clearly establish the relative structure of **2a**. Calculations were performed with the NMR chemical shift calculation protocol equipped on Spartan’18 without changing the default settings (Hehre’s protocol) [43]. The protocol features the chemical shift calculations using ωB97X-D/6-31G*, the evaluation of the energies of individual conformers with ωB97X-V/6-311+G(2df,2p)[6-311G*]//ωB97X-D/6-31G*, and the empirical correction based on the type of carbons, attached atoms, and bond lengths. Some molecules (**1a**, **1b**, **4**, and **5**) were calculated using the *ent*-forms ((11*S*)-enantiomers) of the natural products because the chirality investigation was performed later. However, in this article, their NMR and electronic circular dichroism (ECD) properties are discussed after interpretation that the natural products are all (11*R*)-enantiomers for convenience. In the statistical analysis of ^1^H NMR signals, the methylene protons of the proposed diastereomers were arranged according to the observed NOEs except for unassigned protons such as H_2_-12 of **1a**–**7** and H_2_-15 of **3**, which were set to reduce the chemical shift difference with the calculated values. Similarly, all prochiral nuclei in other diastereomers were arranged so that the chemical shift difference with the calculated values was smaller. Note that the signals were arranged so that the incorrect structures became rather advantageous in the analysis. The calculated ^1^H and ^13^C chemical shifts were directly subjected to statistical and DP4 analysis without empirical correction [44], since Hehre’s protocol involved more sophisticated corrections [43]. Although this protocol has been proved to be more accurate than Goodman’s method [39], Goodman’s standard deviation (*σ*: 2.306 and 0.185 ppm for δ^13^C and δ^1^H, respectively) and freedom (*ν*: 11.38 and 14.18 for δ^13^C and δ^1^H, respectively) were applied in the DP4 analysis [44] because these parameters have not been published for Hehre’s protocol. Accordingly, the DP4 analysis in the present study tended to be slightly less sensitive toward structural differences than that using the appropriate parameters (SI-98-122).

Table 2 summarizes the results. The proposed structures of **1a**–**8** afforded satisfying small root-mean-square of the deviation (RMSD) values for δ^13^C and δ^1^H from the experimental data. The δ^13^C DP4 probability for **6** was not the highest (47.9%), and the 11-epimer gave the best score (52.1%). However, the δ^13^C RMSD values for these isomers were small enough (1.1 and 1.0 ppm, respectively) when considering the average error of this protocol (2.0 ppm) [43]. In other words, the small difference of the calculated δ^13^C values between **6** and its 11-epimer hindered their differentiation. In contrast, the δ^1^H DP4 probability for **6** was quite high (98.5%), which compensated for the unsatisfactory results of the δ^13^C DP4 analysis. Although the relative configuration of **2a** based on the spectroscopic analysis remained unclear, both δ^1^H and δ^13^C DP4 scores strongly supported the proposed structure (δ^13^C: 87.3%, δ^1^H: 99.9%, δ^13^C + δ^1^H: 100%). The ^13^C + ^1^H DP4 values were above 99% probability in all cases. It is worth noting that the δ^13^C DP4 probability of **8** was the highest and exclusive (99.9%) among the 32 possible diastereomers. As described above, these chemical shift calculations not only supported the relative structures of **1a**–**8** obtained by the spectral analysis but also demonstrated the efficiency of this methodology.

The calculation results must be discussed more holistically. The δ^1^H DP4 values generally gave higher scores than the δ^13^C DP4 values for the correct diastereomers of **1a**–**7**. The ^13^C chemical shift is known to depend on parameters such as the electronegativity of neighboring functional groups, bond angles, and bond lengths. Since the substituents methyl and hydroxymethyl groups at C-11 are located at the sterically less hindered end of the molecules, the effect of its geometrical difference on the conformations of the other moieties in **1a**–**7** and on the δ^13^C values can be considered insignificant. In contrast, the anisotropic effect of 12-OH significantly affects the magnetic shielding of the nearby hydrogen atoms. This effect is small for carbon nuclei probably because carbons are located inside the molecules and are usually shielded by hydrogen atoms. This would explain why only the ^1^H chemical shifts are sensitive to the geometry of 12-OH. Despite the high δ^13^C DP4 score obtained for **8**, its δ^1^H DP4 is less conclusive (30.7%). This result is usual in calculations of this type and demonstrates the difficulty of the stereostructural elucidation only based on δ^1^H DP4 values.

Then, the absolute configurations of the natural products were investigated. Tricocerapicanol A (**1a**) provided a characteristic negative Cotton effect (Δε −5.7) at 297 nm attributed to the *R*-band (*n*→π∗ transition) of the C-5 carbonyl group in the ECD spectrum (Figure 3, spectra **A**). This was reproduced by a B3LYP/def2-TZVP model [45,46] when the (11*R*)-enantiomer of **1** was applied (Figure 3, spectra **C**, SI-124). However, the calculated and experimental spectra were considerably different in the 200–230 nm region. The Cotton effect in this region depended on the individual stable conformers (Figure 3, spectra **D**), whereas the negative Cotton effects at approximately 300 nm were constant regardless of the conformers, suggesting that the Cotton effect at approximately 300 nm is more reliable for the chirality assessment than that at approximately 200 nm. It is known that the octant rule can be applied for Cotton effects at the *R*-band of the ketone group [47]. The bicyclo[1–3]-8-octanone framework in **1a** is highly symmetric; thus, this moiety is not responsible for the Cotton effect. Similarly, the Cotton effects of the 14- and 15-methyl groups should cancel each other’s the Cotton effects. Meanwhile, Hα-1 (highlighted with a green sphere in the 3D and 2D models in Figure 3), which is located at the front right lower octant, could be expected to contribute to the negative Cotton effect at approximately 300 nm. This assumption was verified with the simplified virtual models **I** (the model without 12-CH_2_OH and 13-CH_3_), **II** (the model in which the Hα-1 of model **I** was replaced with a chlorine atom), and **III** (the Δ^1,12^ model); the calculations of model **I** afforded a negative Cotton effect at approximately 300 nm similar to that of the experimental spectrum of **1a**, and the intensity of the negative Cotton effect increased in model **II** because of the presence of a polarized C–Cl bond there, whereas that of model **III** was inversed because of the absence of Hα-1 (see SI-145), according to the calculations (SI-145, 146, and 147).

In contrast, **1b** and **1c** showed only faint, negative Cotton effects at approximately 300 nm (Figure 3, spectra **A**), which were moderately well reproduced by the calculations (Figure 3, spectra **C**, SI-126–131). Consequently, the 5-OH group of **1b**, located at the rear left upper octant, contributes to the positive Cotton effect and cancels the negative Cotton effect observed in **1a**. Although the reason for this observation remains unclear, the 14-OH group of **1c** likely weakens the negative Cotton effect observed in **1a**. Note that the DFT calculations can describe these effects more quantitatively. On the basis of these results, it can be concluded that **1a**, **1b**, and **1c** possess the (11*R*)-configuration.

Violascensol (**2b**) was first isolated by Vidari’s group in 1998 and was reported to show a negative Cotton effect at 332 nm [30]. However, its absolute configuration could not be determined on the basis of this observation and was tentatively assigned as 11*R* for **2b** according to its structural and biosynthetic resemblance to related natural products. In 2002, Ferlek mentioned that the general helicity rule for cisoidal enones is not applicable to **2b** in their review discussing the chirotopical properties of cisoidal enones [32]. Both **2a** and **2b** exhibited negative Cotton effects at approximately 335 nm, which were in accordance with Vidari’s report (Figure 4, spectra **A** and **B**), along with positive and negative Cotton effects at around 260 and 220 nm, respectively. All these Cotton effects were nicely reproduced for the (11*R*)-enantiomers of both **2a** and **2b** using a B3LYP/def2-TZVP model (SI-134–137). Interestingly, the simplified virtual model **IV** consisting of the substructure highlighted in red in **2a** and **2b** (Figure 4) also well reproduced the experimental ECD spectrum when the dihedral angle ∠O/C-5/C-6/C-7 was set to 15° (SI-148 and 149), which was nearly identical to that of the stable conformer of **2a** (the numbering followed that used for **2a**). Consequently, the three Cotton effects observed in **2a** and **2b** are most likely caused by the chiral torsion of their cisoidal α,β-unsaturated ketone moiety.

Compounds **3**, **4**, and **5** weakly absorbed UV light at approximately 200 nm, which can be attributed to the *K*-band (π→π* transition) of their isolated double bonds (Figure 5, spectra **A**–**C**). Coincidentally, all these compounds showed positive Cotton effects at this wavelength region in their ECD spectra, which were well reproduced when the (11*R*)-enantiomers of **3**–**5** were subjected to ECD calculations (SI-136–142). Although these compounds involved more than 30 stable conformers within 10 kJ/mol from the global minimum, the 10 most stable conformers showed positive Cotton effects at around 210 nm in all cases (spectra **D**–**F**), which confirms the reliability of the above argument. In the UV/ECD spectral reproductions for **3**, and the BHLYP functional provided a better match than B3LYP (SI-152), whereas the latter functional afforded satisfying UV/ECD spectra for other compounds. This highlights the necessity of using various functionals for a reliable elucidation of the chirality. The suggested chirality of **3** was consistent with that of **2a** and **2b**, which was expected from the viewpoint of their biosynthesis. For **3**–**5**, attempts at applying Scott’s empirical octant rule for olefins [48] were not conclusive.

The absolute configuration of omphadiol (**8**) was established to be the (10*R*)-enantiomer in the enantioselective total synthesis reported by Romo [33]. Nevertheless, its 10-*O*-benzoate **8-OBz** was independently investigated in the present study. Natural **8** showed no remarkable Cotton effects at the UV/vis region, whereas **8-OBz** showed a characteristic ECD spectrum, as shown in Figure 6. Such complexed Cotton effects most likely stem from the interaction of the π character of the cyclopropane ring [49] with the benzoyl chromophore. The experimental ECD spectrum was successfully reproduced by the DFT B3LYP/def2-TZVP functional (SI-143 and 144), confirming the (10*R*)-configuration. Interestingly, the experimental ECD spectrum of **8-OBz** could be roughly reproduced with the simplified virtual model **V**, which only contains the substructure highlighted in red in **8-OBz** (Figure 6), when the dihedral angle ∠C-7/C-9/C-10/O was set to 170°, which is the angle of the corresponding atoms in the stable conformation of **8** (SI-150 and 151). These demonstrated that the DFT-based modeling calculation is an efficient tool not only for chirality determinations but also for revealing the major factors causing the Cotton effects.

Although tricoilludins B (**6**) and C (**7**) do not possess either appropriate functionals or chromophores for chirality elucidation, the (11*R*)-configuration could be assigned by considering that they were isolated from the same culture broth as **5**.

These molecules belong to the cyclohumulanoid family, and except for **8**, contain a C-1/C-2/C-9/C-10/C-11 cyclopentane ring sharing methyl (C-13) and hydroxymethyl (C-12) groups at C-11 with (*R*)-configuration, although the absolute configurations of **6** and **7** were not unequivocally established. A protoilludane derivative **10** can be envisaged as the key biosynthetic intermediate for the compounds obtained in the present study. Protoilludane **10** is likely derived from farnesyl pyrophosphate via humulenyl cation **9** as shown in Scheme 1. According to the established configurations, the oxidation of C-12 occurs stereoselectively right after cyclization because all compounds except for **8** contain an (11*R*)-hydroxymethyl group. Humulenyl cation **9** also produces **8** in another tandem cyclization at C-7/C-9 and C-2/C-6; however, a different type of enzyme may be responsible for this process [8,10]. Protoilludanes **2a**, **2b**, and **3** are obtained by oxidation(s) of **10**. Epoxide **11** can be proposed as a versatile intermediate yielding cerapicanes **1a**–**c**, sterpurane **4**, and illudanes **5**–**7**. An epoxide ring opening reaction at C-6 induces a ring contraction of cyclobutene into a cyclopropane, affording illudanes **5–7**. Another epoxide ring opening at C-7 promotes a ring expansion into cyclopentane, generating cerapicanium cation **12**. Deprotonation at 5-OH affords ceapicanes **1a**–**1c**, whereas another deprotonation at H-2 produces a concerted olefin formation and ring contraction into cyclobutane to give tricosterpurol (**4**).

Finally, an antifungal assay was performed using *Cochliobolus miyabeanus* as the test fungus to reveal that **2a** and **5** simultaneously produced swelling and branching at a concentration of 5 μg/mL (SI-153), whereas other compounds did not show obvious inhibitory effect even at higher concentrations.

## 3. Experimental

**General Experimental Procedures.** UV spectra were obtained on a HITACHI U-2010 spectrometer (Hitachi High-Tech, Tokyo, Japan) using a 10 mm length cell. ECD spectra were recorded on a JASCO J-1100 spectropolarimeter (JASCO Cooperation, Tokyo, Japan) with a 10 mm length cell. Fourier transform infrared spectroscopy was conducted using a HORIBA FT-720 spectrometer (Horiba Ltd, Kyoto, Japan) and a KBr cell. ^1^H and ^13^C NMR spectra were recorded on a JEOL JNM-ECX500 spectrometer (^1^H: 500 MHz, ^13^C: 125 MHz, (JEOL Ltd., Tokyo, Japan). Tetramethylsilane (0 ppm) was used as the internal standard for both types of spectra when CDCl_3_ was used as the solvent. When acetone-*d*_6_ was used, the signals of CHD_2_COCD_3_ (2.05 ppm) and ^13^CD_3_COCD_3_ (29.92 ppm) were used as the internal standard for the ^1^H and ^13^C NMR spectra, respectively. ESI-TOFMS spectra were obtained using a HITACHI NanoFrontier LD spectrometer equipped with a HITACHI 2100 high-performance liquid chromatography (HPLC) pump, a HITACHI L-2420 UV detector (Hitachi High-Tech, Tokyo, Japan), and a HITACHI L-2300 column oven. Calibration was performed with a mixture of tetrabutylammonium ion (*m*/*z* 242.2848), reserpine (*m*/*z* 609.2807), and Ultramark 1621. Silica gel thin-layer chromatography (TLC) analyses were conducted using Merck TLC silica gel 60 F254 plates (No. 5715) (Merck & Co., Kenilworth, NJ, USA). Silica gel column chromatography was performed using silica gel Merck 707734. Chemicals and solvents were purchased from FUJIFILM Wako Pure Chemical Cooperation and Sigma-Aldrich Co. LLC (St. Louis, MO, USA) and used without further purification. Conformation searches and chemical shift calculations were performed with Spartan’18 (Wavefunction, Irvine, CA, USA) using a PC (operating system: Windows 7 Professional; CPU: Intel Xeon E5-2697 v2 processor, 2.70 GHz, 12 cores; RAM: 128 GB). ECD spectra were calculated using TmoleX 2021 (Dassault Systèms, Vélizy-Villacoublay, France) on a PC workstation (operating system: CentOS 7.1.1; CPU: Intel Xeon E5-2687W V4, 3.0 GHz, 12 cores × 2; RAM: 256 GB). The calculated ECD spectra were constructed using Microsoft Excel for Microsoft Office365 on a commercial PC (Windows 10, Microsoft, Redomond, WA, USA).

**Fungus.***D*. *tricolor* was sampled in 2019 at the Shirakami Natural Science Park of Hirosaki University located in the Shirakami Mountains, Japan.

**Isolation.***D*. *tricolor* was cultured in a potato dextrose medium (200 mL in a 500 mL baffled Erlenmeyer flask × 45) on a rotary shaker (110 rpm) at 26 °C for 14 days. MeOH (150 mL) was then added to each flask to precipitate the fungus body. After filtration through cotton gauze, the MeOH of the combined filtrate was removed with a rotary evaporator. The resulting aqueous suspension was extracted with ethyl acetate (3 L × 3), and the organic layer was combined and concentrated under vacuum conditions to obtain the crude extracts (1.7 g). These operations were repeated to obtain the second crude extracts (1.6 g). The first crude extracts were diluted with EtOAc (100 mL), and then, silica gel (~6.0 g) was added. The resulting suspension was carefully concentrated under reduced pressure. The obtained residual powder was loaded on a silica gel column (300 g) and eluted with 0→100% EtOAc/hexane to give fraction A to fraction G. Fraction C eluted with 20–30% EtOAc/hexane in the first column chromatography (113 mg) was further subjected to ODS medium-pressure column chromatography (YAMAZENE ODS universal column type S, 50–100% MeOH/H_2_O for 20 min, flow rate 5 mL/min) to give **1a** (11.6 mg). Fraction D eluted with 50–60% EtOAc/hexane in the first column chromatography (205 mg) was similarly subjected to the ODS medium-pressure column chromatography (YAMAZENE ODS universal column type S, 50–100% MeOH/H_2_O for 30 min, flow rate 5 mL/min) to give fractions D-1 to D-4. Fraction D-2 (35.0 mg) was subjected to a preparative silica gel TLC (30% EtOAc/hexane, two developments) to give **8** (1.5 mg, *R_f_* = 0.3), **2b** (1.8 mg, *R_f_* = 0.45), and **2a** (1.0 mg, *R_f_* = 0.2). Fraction E eluted with 70–80% EtOAc/hexane in the first column chromatography (186 mg) was further subjected to ODS medium-pressure column chromatography (YAMAZENE ODS universal column type S, 50–100% MeOH/H_2_O for 20 min, flow rate 5 mL/min) to give crude **1b** (40 mg). Analytical **1b** (10 mg) was obtained by preparative silica gel chromatography (70% EtOAc/hexane). The second crude extracts were also dispersed in silica gel (6.0 g) and subjected to silica gel column chromatography as described above to obtain fraction A′ to fraction J′. Similar separations afforded **1a** (13.4 mg) and **2a** (1.4 mg). Fraction A′ eluted with 10% EtOAc/hexane (8.0 mg) was subjected to preparative silica gel chromatography (30% EtOAc/hexane) to obtain **5** (5.5 mg, *R_f_* = 0.3). Fraction J′ eluted with 80% EtOAc/hexane (434 mg) was dispersed on diatomaceous earth powder (1.0 g) in MeOH (20 mL), and the solvent was then carefully evaporated under reduced pressure. The obtained dried powder was placed in a small column and connected to a YAMAZENE ODS universal column type L, which was developed using gradient conditions (5→100% MeOH/H_2_O (containing 1% AcOH) for 250 min, flow rate 3 mL/min). The eluents were fractionated into 85 test tubes. Each fraction was checked with silica gel TLC and integrated into eight fractions (J′-1 to J′-8). Fraction J′-2 eluted with 10–15% MeOH/H_2_O (22.7 mg) was further subjected to silica gel column chromatography (~5 g, 50% EtOAc/hexane) to yield **6** (1.2 mg).

Fraction J′-6 eluted with 40–50% MeOH/H_2_O (91.8 mg) was subjected to ODS medium-pressure chromatography (YAMAZENE ODS universal column type M, 20→60% MeOH/H_2_O for 60 min, flow rate 8 mL/min). The fractions eluted with 30% and 35% MeOH/H_2_O were recovered to obtain **7** (3.6 mg) and crude **3** (24.3 mg), respectively. The latter fraction was subjected to ODS HPLC (Wakopak^®^ Ultra C18-5; φ 20 mm × 250 mm, 20% CH_3_CN/H_2_O (containing 0.1% TFA), flow rate 10 mL/min, detected at 214 nm) to yield **3** (10.9 mg, *t*_R_ = 28.0 min). Fraction J′-7 eluted with 50–60% MeOH/H_2_O (115 mg) was subjected to silica gel column chromatography (~5 g, 30% EtOAc/hexane) to yield **4** (3.9 mg). Fraction J′-8 eluted with 70–80% MeOH/H_2_O (25.1 mg) was subjected to ODS HPLC (Wakopak^®^ Ultra C18-5; φ 20 mm × 250 mm, 20→40% CH_3_CN/H_2_O (containing 0.1% TFA) for 30 min, flow rate 10 mL/min, detected at 220 nm) to give **1c** (1.4 mg, *t*_R_ = 22.7 min). 

**Physical data of 1a:** ECD (2.85 × 10^−4^ mol/L in CH_3_CN): 297 nm (Δε −5.7); UV (2.85 × 10^−4^ mol/L in CH_3_CN): 236 nm (sh, ε 760), 196 nm (ε 2,800); IR (film) 3400, 2954, 2927, 2868, 1733, 1455, 1029 cm^−1^; ^1^H NMR and ^13^C NMR data in CDCl_3_ are shown in Table 1. ESI-TOFMS (rel. int (%), assignment) *m*/*z* 254.2139 (35, [M + NH_4_]^+^: 254.2115), 237.1869 (100, [M + H]^+^: 237.1849).

**Physical data of 1b:** ECD (7.0 × 10^−4^ mol/L in CH_3_CN) 295 nm (Δε −0.3); UV (7.0 × 10^−4^ mol/L in CH_3_CN) 230 (sh, ε 310), 196 nm (ε 1600); IR (film) 3450, 2930, 2867, 1732, 1450, 1045 cm^−1^; ^1^H NMR and ^13^C NMR data in CDCl_3_ are shown in Table 1; ESI-TOFMS (rel. int (%), assignment) *m*/*z* 527.3369 (45, [2 M + Na]^+^: 527.3343), 270.2065 (50, [M + Na]^+^: 270.1618), 253.1802 (100, [M + H-H_2_O]^+^: 253.1798).

**Physical data of 1c:** ECD (2.78 × 10^−4^ mol/L in CH_3_CN) 300 nm (Δε −0.8); UV (2.78 × 10^−4^ mol/L in CH_3_CN) 230 (sh, ε 310), 195 nm (ε 2500); IR (film) 3365, 2927, 2870, 1732, 1043 cm^−1^; ^1^H NMR and ^13^C NMR data in CDCl_3_ are shown in Table 1; ESI-TOFMS (rel. int (%), assignment) *m*/*z* 527.3374 (12, [2M + Na]^+^: 527.3343), 253.1802 (100, [M + H-H_2_O]^+^: 253.1798).

**Physical data of 2a:** ECD (2.82 × 10 ^−4^ mol/L in CH_3_CN) 335 nm (Δε −0.4); 261 nm (Δε +1.0), 224 nm (Δε −1.5); UV (2.82 × 10 ^−4^ mol/L in CH_3_CN) 258 nm (ε 1720); IR (film) 2956, 2923, 1737, 1675 cm^−1^; ^1^H NMR and ^13^C NMR data in CDCl_3_ are shown in Table 1; ESI-TOFMS (rel. int (%), assignment) *m*/*z* 519.2747 (30, [2M + Na]^+^: 519.2717), 271.1297 (20, [M + Na]^+^: 271.1305), 249.1482 (100, [M + H]^+^: 249.1485).

**Physical data of 2b:** ECD (6.35 × 10^−4^ mol/L in CH_3_CN) 333 nm (Δε −0.4), 258 nm (Δε +0.8), and 225 nm (Δε −1.0); UV (6.35 × 10^−4^ mol/L in CH_3_CN) 258 nm (ε 1140); IR (film) 3425, 2925, 2867, 1728, 1668, 1043 cm^−1^; ^1^H NMR (500 MHz, CDCl_3_) δ 1.06 (1H, brd, *J* = 12.2 Hz, H_2_-10), 1.13 (3H, s, H_3_-13), 1.18 (3H, s, H_3_-14), 1.43 (1H, dd, *J* = 9.3, 13.5 Hz, H_2_-1), 1.43 (1H, dd, *J* = 9.8, 13.6 Hz, H_2_-8), 1.73 (1H, ddd, *J* = 1.5, 8.8, 13.5 Hz, H_2_-1), 1.92 (1H, ddd, *J* = 1.0, 7.3, 12.2 Hz, H_2_-10), 2.01 (3H, s, H_3_-15), 2.18 (1H, ddd, *J* = 8.8, 9.3, 13.5 Hz, H-2), 2.25 (1H, dq, *J* = 6.5, 14.9 Hz, H_2_-8), 2.41 (1H, m, H-9), 2.66 (1H, d, *J* = 16.8 Hz, H_2_-4), 2.69 (1H, d, *J* = 16.8 Hz, H_2_-4), 3.40 (1H, d, *J* = 10.7 Hz, H_2_-12), 3.43 (1H, d, *J* = 10.7 Hz, H_2_-12). ^13^C NMR (CDCl_3_) δ 20.4 (C-14), 20.4 (C-15), 24.5 (C-13), 35.8 (C-8), 36.2 (C-1), 36.8 (C-3), 41.8 (C-9), 43.6 (C-10), 45.5 (C-11), 47.4 (C-2), 60.9 (C-4), 69.4 (C-12), 143.3 (C-7), 150.7 (C-6), 197.2 (C-5). The spectral data are consistent with those in the literature [30]. ESI-TOFMS (rel. int (%), assignment) *m*/*z* 491.3080 (30, [2M + Na]^+^: 491.3132), 469.3252 (35, [2M + H]^+^: 469.3312), 235.1666 (100, [M + H]^+^: 235.1693).

**Physical data of 3:** ECD (6.35 × 10^−4^ mol/L in CH_3_CN) 214 nm (ε +1.3); UV (6.35 × 10^−4^ mol/L in CH_3_CN); 205 nm (ε 7500); IR (film) 3350, 2925, 2870, 1463, 1040 cm^−1^; ^1^H NMR and ^13^C NMR data in CDCl_3_ are shown in Table 1; ESI-TOFMS (rel. int (%), assignment) *m*/*z* 527.3391 (15, [2M + Na]^+^: 527.3343), 275.1620 (100, [M + Na]^+^: 275.1618).

**Physical data of 4:** ECD (1.06 × 10^−4^ mol/L in CH_3_CN) 206 nm (Δε +10.9); UV (1.06 × 10^−4^ mol/L in CH_3_CN) 202 nm (ε 8100); IR (film) 3310, 2920, 2863, 1455, 1040 cm^−1^; ^1^H NMR and ^13^C NMR data in CDCl_3_ are shown in Table 1; ESI-TOFMS (rel. int (%), assignment) *m*/*z* 219.1760 (100, [M + H-H_2_O]^+^: 219.1743), 201.1653 (55, [M + H-2H_2_O]^+^: 201.1638).

**Physical data of 5:** ECD (1.31 × 10^−4^ mol/L in CH_3_CN) 212 nm (ε +5.3); UV (1.31 × 10^−4^ mol/L in CH_3_CN) 203 nm (ε 8600). IR (film) 3394, 2954, 2850 cm^−1^. ^1^H NMR and ^13^C NMR data in CDCl_3_ are shown in Table 1. ESI-TOFMS (rel. int (%), assignment) *m*/*z* 219.1738 (100, [M + H-H_2_O]^+^: 219.1743), 201.1633 (45, [M + H-2H_2_O]^+^: 201.1638).

**Physical data of 6:** This compound showed no considerable absorption in the UV region. IR (film) 3400, 2923, 2869, 1025 cm^−1^; the ^1^H and ^13^C NMR data in CDCl_3_ are shown in Table 1; ESI-TOFMS (rel. int (%), assignment) *m*/*z* 275.1623 (45, [M + Na]^+^: 275.1623), 235.1700 (50, [M + H-H_2_O]^+^: 235.1693), 217.1592 (65, [M + H-2H_2_O]^+^: 217.1587), 167.0133 (100, not assigned).

**Acetylation of 6:** A solution of **6** (~0.5 mg) in pyridine (0.3 mL) was stirred with acetic anhydride (0.1 mL) at room temperature for 5 h. The reaction mixture was concentrated under reduced pressure, and the resulting residue was subjected to preparative TLC (*R_f_* = 0.5, 80% EtOAc/hexane) to obtain the 12-*O*-acetate of **6**. NMR spectra were measured using a SHIGEMI symmetrical MICRO NMR tube (SHIGEMI CO., LTD, Tokyo, Japan). IR (film) 3446, 2923, 2850, 1733 cm^−1^, ^1^H NMR (500 MHz, CDCl3) δ 0.30 (1H, dt, *J* = 10.0, 5.7 Hz), 0.52 (1H, dt, *J* = 5.7, 10.0 Hz), 0.66 (2H, overlapped), 0.94 and 1.16 (each 3H, s), 1.50–1.72 (5H, overlapped), 1.81 (1H, dd, *J* = 9.5, 14.7 Hz), 2.08 (3H, s), 2.33 (1H, m), 2.46 (1H, m), 3.47 (1H, dd, *J* = 1.9, 7.9 Hz), 3.78 and 3.90 (each 1H, AB doublet, *J* = 10.7 Hz), 3.95 (1H, d, *J* = 7.9 Hz), ^13^C NMR (125 MHz, CDCl_3_) d 2.3, 3.2, 21.0, 21.7, 24.8, 34.2, 35.3, 37.6, 39.1, 41.5, 43.9, 46.4, 70.8, 71.2, 78.0, 83.0, 171.5; ESI-TOFMS (rel. int (%), assignment) *m*/*z* 317.1737 (100, [M + Na]^+^: 317.1723). 

**Physical data of 7:** This compound afforded no considerable absorption in the UV region. IR (film) 3370, 2930, 2870, 1455, 1478, 1022 cm^−1^; the ^1^H and ^13^C NMR data in CDCl_3_ are shown in Table 1; ESI-TOFMS (rel. int (%), assignment) *m*/*z* 253.1793 (25, [M + H]^+^: 253.1798), 235.1689 (100, [M+H-H_2_O]^+^: 235.1693), 217.1580 (65, [M + H-2H_2_O]^+^: 217.1587). 

**Physical data of 8:** ^1^H NMR (500 MHz, CDCl_3_) δ 0.45 (1H, t, *J* = 4.5 Hz, H-8), 0.55 (1H, dt, *J* = 5.8, 8.4 Hz, H-9), 0.74 (1H, dt, *J* = 8.0, 4.5 Hz, H-8), 0.97 (3H, s, H_3_-12), 0.99 (3H, s, H_3_-15), 1.02 (3H, s, H_3_-12), 1.19 (t, *J* = 12.6 Hz, H-1), 1.27 (3H, s, H_3_-14), 1.42 (1H, overlapped, H-6), 1.45 (1H, dd, *J* = 2.5, 12.6 Hz, H-1), 1.60 (1H, overlapped, H-4), 1.61 (1H, overlapped, H-2), 1.70 (2H, overlapped, H-4, H-5), 1.80 (overlapped, H-5), 3.14 (1H, d, *J* = 8.0 Hz, H-10), ^13^C NMR (125 MHz, CDCl_3_) δ 19.1 (C-12), 19.4 (C-15), 19.5 (C-7), 22.6 (C-8), 23.2 (C-5), 25.7 (C-14), 28.7 (C-13), 29.7 (C-9), 38.0 (C-11), 41.4 (C-4), 42.1 (C-1), 48.2 (C-2), 49.6 (C-6), 81.0 (C-10), 81.1 (C-3). The spectral data are consistent with those in the literature [31]. ESI-TOFMS did not provide considerable ions.

**Benzoylation of 8****:** A solution of **8** (~1.0 mg) in pyridine (0.1 mL) was stirred with benzoyl chloride (2.0 μL) and 4-(dimethylamino)pyridine (1.0 mg) at room temperature for 30 min. MeOH (1.0 mL) was then added, and the mixture was stirred at room temperature. After 1 min, diethylether (1.0 mL) was added, the resulting suspension was filtered through cotton, and the filtrate was concentrated under reduced pressure. Preparative silica gel TLC of the residue (30% EtOAc/hexane) afforded the 10-*O*-benzoate of **8** (~1.1 mg). The yield of this compound was estimated according to the UV absorption at 230 nm by assuming an ε value of 15,300. ECD 4.66 × 10^−5^ mol/L (CH_3_CN) 239 nm (Δε +1.7) and 206 nm (Δε +1.0), UV 229 nm (ε 15300). IR (film) 3506, 3433, 2958, 2925, 2852, 1714, 1277, 1115 cm^−1^. ^1^H NMR (CDCl_3_) δ 0.71 (1H, H-9), 0.73 (2H, H_2_-8), 0.97 (3H, s, H_3_-13), 1.02 (3H, s, H_3_-15), 1.16 (3H, s, H_3_-12), 1.29 (3H, s, H_3_-14), 1.36 (1H, t, *J* = 7.8 Hz, H-4), 1.51 (1H, H_2_-1), 1.61 (1H, H-2), 1.62 (1H, H-6), 1.71 (1H, H-4), 1.73 (1H, H-5), 1.88 (1H, m, H-5), 4.77 (1H, d, *J* = 8.1 Hz, H-10), 7.44 (2H, aromatic protons), 7.55 (2H, aromatic protons), 8.07 (1H, aromatic protons). ^13^C NMR (CDCl_3_) δ 19.4 (C-15), 19.6 (C-7), 20.5 (C-12), 23.2 (C-5), 23.3 (C-8), 25.7 (C-14), 27.1 (C-9), 28.7 (C-13), 37.9 (C-11), 41.5 (C-4), 41.9 (C-1), 48.1 (C-6), 49.3 (C-2), 81.0 (C-3), 82.7 (C-10), 128.3, 129.6, 131.0, 132.7 (aromatic carbons), 165.7 (C=O). ESI-TOFMS (rel. int (%), assignment) *m/z* 360.2539 (25, [M + NH_4_]^+^: 360.2533), 203.1787 (100, [M + H–OBz–H_2_O]^+^: 203.1794).

**Chemical shift calculations.** (11*S*)-**1a**, (11*S*)-**1b**, (11*R*)-**1c**, (11*R*)-**2a**, (11*R*)-**2b**, (11*R*)-**3**, (11*S*)-**4**, (11*S*)-**5**, (11*R*)-**6**, (11*R*)-**7,** (10*R*)-**8** and their possible diastereomers were built on Spartan’18 and were directly subjected to the chemical shift calculation protocol with a default setting of the program [39], which automatically performed the conformational search with MMFF, followed by the collection of the candidate conformers by setting the threshold at 40 kJ/mol from the global minimum conformer; structure re-optimization employing the HF/3-21G model, followed by conformer narrowing by setting the threshold at 40 kJ/mol from the global minimum conformer; energy estimation using the ωB97X-D/6-31G* model, followed by conformer narrowing by setting the threshold at 15 kJ/mol from the global minimum conformer; structural optimization with the ωB97X-D/6-31G* model, followed by conformer narrowing by setting the threshold at 10 kJ/mol from the global minimum conformer; energy estimation applying the ωB97X-V/6-311+G(2df,2p)[6-311G*] model, followed by conformer narrowing by setting the threshold at 10 kJ/mol from the global minimum conformer; chemical shift calculations using ωB97X-D/6-31G*, followed by the empirical correction [39]. The obtained chemical shifts were directly compared with the experimental data, and the result data (RMSD, maximum deviation, and DP4) were calculated using Microsoft Excel for Microsoft 365. The methylene protons of the proposed diastereomers except for H-12 were arranged according to the assignments on the basis of the NOE analysis, and H_2_-12 and all the methylene signals in other isomers were set to reduce the chemical shift difference with the calculated values. The obtained ^1^H and ^13^C chemical shifts were directly subjected to statistical analysis without corrections on the basis of the slopes and intercepts. In the DP4 analysis, the original parameters provided by Goodman’s method (*σ*: 2.306 ppm for ^13^C and 0.185 ppm for ^1^H; *ν*: 11.38 for ^13^C and 14.18 for ^1^H) were used [44]. 

**ECD spectral calculations.** The stable conformer sets obtained in the chemical shift calculations were further optimized with B3LYP/def2-TZVP on TmoleX 2021. Only the most 10 stable conformers were subjected to the ECD calculations when more than 10 stable conformers within 10 kJ/mol from the global minimum conformers were obtained with ωB97X-V/6-311+G(2df,2p)[6-311G*//ωB97X-D/6-31G* (**1b**, **1c**, **3**, **4**, **5**, **6**, and **7**). After vibrational analysis, UV/ECD calculations were performed with B3LYP/def2-TZVP. The UV and ECD spectra of the individual conformers were constructed according to the frequencies, oscillator strength, and rotatory strengths using the NORM.DIST function in Microsoft Excel for Microsoft 365. The widths of the UV and ECD bands were set to reproduce the spectra appropriately. The UV/ECD spectra of the individual conformers were constructed by Boltzmann averaging on the basis of the free energy. The wavelengths of the UV spectra were corrected according to the experimental spectra, and those of the ECD spectra were corrected using the identical number. The ECD spectra of **1a**, **1b**, **4**, and **5** were expressed as the values obtained after multiplying by −1.

Model **I** was constructed on Spartan’18 by removing nonessential atoms from the most stable conformer of **1a**. For model **II**, Hα-1 of model **I** was replaced with chlorine. Model **III** was prepared from model **I** by introducing a double bond between C-1 and C-11. The resulting structures were then optimized with B3LYP/def2-TZVP and subjected to ECD calculations as described above. The most stable conformation of **8-OBz** was obtained by performing a conformational search with MMFF, and the 6 stable conformers were subjected to the ECD calculations in a similar manner to that above. Model **IV** and model **V** were prepared by removing nonessential atoms from the most stable conformations of **2a** and **8-OBz**, respectively. The structures obtained by setting the dihedral angles ∠O/C-5/C-6/C-7 in model **IV** and ∠C-7/C-9/C-10/O in model **V** to 15° and 170°, respectively, were optimized with B3LYP/def2-TZVP by fixing the set torsions and then subjected to UV/ECD calculations. The calculated UV/ECD spectra were constructed similarly to those above. 

**Antifungal assay.** A series of suspensions of spores of *Cochliobolus miyabeanus* (1.0 mL) containing 1, 5, 25, and 100 μg/mL of samples in Petri dishes were prepared and incubated at 25 °C for 24 h. Then, microscopic observation was performed using an Olympus CKX-41 binocular inverted microscope equipped with a ×10 objective lens and ×10 eyepieces.

## 4. Conclusions

In summary, nine cyclohumulanoids were successfully identified from *D*. *tricolor*. The present study proves the usefulness of DFT-based modeling calculations for the determination of the relative and absolute configurations of polycyclic terpenoids and the suitability of calculations using simplified models for elucidating the reasons behind the Cotton effects.

## Data Availability

Most of the data in this article are disclosed in the Appendix A. Further data presented in this study are available on request from the corresponding author.

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
