# Peer review of "Cyclohumulanoid Sesquiterpenes from the Culture Broth of the Basidiomycetous Fungus Daedaleopsis tricolorâ€"

_molecules, 2021, doi:10.3390/molecules26144364_

Round 1
Reviewer 1 Report
This manuscript describes seven new compounds obtained from the mushroom Daedaleopsis tricolor and their bioactivities. The structures were fully elucidated on the basis of NMR spectroscopic analysis, and the suggested relative structures were confirmed via density functional theory (DFT)-based chemical shift calculations involving a DP4 probability analysis. This paper is proper for Molecular after minor revisions.
- Page 2, Line 75 and Line 77: “nuclear Overhauser effects (NOEs)” should be revised as nuclear overhauser effects (NOEs)”.
- Page 3: Table 1, in the position of H-4 and H-8 of compound 3, there are some statistics defected, please check.
- Page 4: Table 1, The similar error of the 1H NMR data of compound 6 with compound 3, please check.
- Page 6 Line 117, there are some misprints on line 117 page 6.
- Pages18-19, there are some formats lack standardization and unification in the references, please check.
Author Response
- Page 2, Line 75 and Line 77: “nuclear Overhauser effects (NOEs)” should be revised as nuclear overhauser effects (NOEs)”.
Overhauser is name (Albert W. Overhauser) who found the effect. Thus "O" should be capitalized.
- Page 3: Table 1, in the position of H-4 and H-8 of compound 3, there are some statistics defected, please check.
Thank you very much. These were fixed.
- Page 4: Table 1, The similar error of the 1H NMR data of compound 6 with compound 3, please check.
Thank you very much. It was fixed.
- Page 6 Line 117, there are some misprints on line 117 page 6.
So far I checked, it seemed fine.
- Pages18-19, there are some formats lack standardization and unification in the references, please check.
The reviewer suggests capitalization of article titles. These were unified.

Reviewer 2 Report
The manuscript "Cyclohumulanoid Sesquiterpenes from the Culture Broth of the Basidiomycetous Fungus Daedaleopsis tricolor" by R. Kanehara et al. is devoted to elucidating the structure of the cyclohumulanoids produced by the fungus D. tricolor that was sampled in the Shirakami Mountains, Japan. As a result of a comprehensive study, it was shown that DFT-based modeling is a sufficiently effective computational method for chirality determining and explaining the Cotton effects for the studied sesquiterpenoids contained a C-1 / C-2 / C-9 / C-10 / C-11 cyclopentane ring sharing C13 methyl and C2 hydroxymethyl groups at C11 atom with (R)-configuration. The absolute configurations of 6 and 7 were not unequivocally established. However, the biosynthetic pathway for the formation of compounds from the common protoylludane precursor proposed by the authors indirectly allows the studied sesquiterpenoids to be assigned to the same group with the (R) -configuration of 12C atoms.
The manuscript is very well and logically organized. At the same time, the authors must make a small clarifications in order for the manuscript to be accepted:
- The title of the manuscript uses an incorrect combination "Cyclohumulanoid Sesquiterpenes ....", which must be refined. Can be replaced by: "Cyclohumulane Sesquiterpenes ..."
- Figure 3A: the number of compound 1 ... is to be replaced with .....1а
- L. 246: ".... of the C-5 carbonyl group"... is to be replaced with ..... "of the C-6 carbonyl group"
- L. 350: "4. experimental" ... is to be replaced with ..... "4. Experimental"
- [alpha]D data should be reported for most of the new compounds. In the case of previously described compounds 2b and 8, the Experimental section should provide the original data of [alpha]D.
- It is necessary to uniformly indicate the titles of publications in the "References" section.
- As a question: Did the authors attempt to obtain crystals of one of the studied cyclohumulanoids? According to Ref. 31, T.C. McMorris et al. succeeded in obtaining crystals of the 3,5-dinitrobenzoyl derivative of omphadiol isolated from the Basidiomycete Omphalotus illudens for X-ray diffraction analysis.
Author Response
- The title of the manuscript uses an incorrect combination "Cyclohumulanoid Sesquiterpenes ....", which must be refined. Can be replaced by: "Cyclohumulane Sesquiterpenes ..."
Cyclohumulane does not exist because humulene is already cyclic. Cyclohumulanoid means a family of polycyclic congeners derived from humulene. So please allow us to keep the present style.
- Figure 3A: the number of compound 1 ... is to be replaced with .....1а.
Thank you for the suggestion. It was fixed.
- L. 246: ".... of the C-5 carbonyl group"... is to be replaced with ..... "of the C-6 carbonyl group"
Thanks, it was my fault and fixed.
- L. 350: "4. experimental" ... is to be replaced with ..... "4. Experimental"
It was my fault and fixed,
- [alpha]D data should be reported for most of the new compounds. In the case of previously described compounds 2b and 8, the Experimental section should provide the original data of [alpha]D.
Since we measured ECD, specific rotation values were not measured. Unfortunately, the most material has been used in biological assay. Please forgive without those values.
- It is necessary to uniformly indicate the titles of publications in the "References" section.
The reviewer suggests capitalization of article titles. These were unified.
- As a question: Did the authors attempt to obtain crystals of one of the studied cyclohumulanoids? According to Ref. 31, T.C. McMorris et al. succeeded in obtaining crystals of the 3,5-dinitrobenzoyl derivative of omphadiol isolated from the Basidiomycete Omphalotus illudens for X-ray diffraction analysis.
So far we handled the materials, these were not crystallized. We have not tried crystallization by way of derivatizations.
